# Nurse Leadership and Work Environment Association with Outcome Expectancy and Self-Efficacy in Evidence-Based Practice among Hospital Nurses in The Netherlands: A Cross-Sectional Study

**DOI:** 10.3390/ijerph192114422

**Published:** 2022-11-03

**Authors:** Peter Hoegen, Mireille Vos, Catharina van Oostveen, Cindy de Bot, Michael A. Echteld, Jolanda Maaskant, Hester Vermeulen

**Affiliations:** 1School of Health and Social Care, Avans University of Applied Science, Hogeschoollaan 1, 4818 CR Breda, The Netherlands; 2Bachelor of Nursing, Inholland University of Applied Sciences, De Boelelaan 1109, 1081 HV Amsterdam, The Netherlands; 3Research Department, Spaarne Gasthuis, Spaarnepoort 1, 2134 TM Hoofddorp, The Netherlands; 4Erasmus School of Health Policy and Management, Erasmus University Rotterdam, Burgemeester Oudlaan 50, 3062 PA Rotterdam, The Netherlands; 5Research and Education in Nursing Consortium (RE-Nurse), Hilvarenbeekseweg 60, 5022 GC Tilburg, The Netherlands; 6Master EBP in Healthcare, University of Amsterdam, Meibergdreef 9, 1105 AZ Amsterdam, The Netherlands; 7Radboud Institute for Health Sciences, Scientific Center for Quality of Healthcare (IQ Healthcare), Radboud University Medical Center, Kapittelweg 54, 6525 EP Nijmegen, The Netherlands

**Keywords:** evidence-based practice, nurses, nursing leadership, nursing management self-efficacy, outcome expectancy, work environment

## Abstract

The active participation of nurses in evidence-based practice (EBP) is challenging and topical, as shown by the worldwide calls for appropriate, accessible, affordable care and the de-implementation of unnecessary care. Nurses’ perceived support from their managers and work environments may affect their self-efficacy and outcome expectancy in EBP, as well as hinder them in EBP. Associations between these issues have not yet been explored. This study examines the association of self-efficacy and outcome expectancy levels in EBP, as well as the perceived support for EBP from nurse leaders and in the working environment, among Dutch hospital nurses. Methods. Questionnaires measuring nurses’ self-efficacy, outcome expectancy, and perceived support for EBP from nurse leaders and their work environment were completed by 306 nurses in eight hospitals between March 2021 and June 2021. We used multilevel regression analyses to determine the associations and covariates. Results. This study shows that EBP-supportive leaders and work environments positively contribute to nurses’ self-efficacy and outcome expectancy in EBP, along with the covariates undertaking EBP activities and educational level. Conclusions. To improve nurses’ active participation in EBP, nurses need to increase their self-efficacy and outcome expectancy in EBP. Supportive leaders and a supportive work environment do have an impact. Hence, these factors need attention when implementing EBP among nurses.

## 1. Introduction

Evidence-based practice (EBP) is the joint, mutually informed decision-making in healthcare situations, based on weighing arguments from patients’ knowledge, values, and preferences and the best, most current scientific and professional insights, resulting in cost-effective care and better outcomes for individual patients [1,2,3]. Of all hospital staff, nurses spend the most time in direct patient care, which is challenging considering the current calls for appropriate, accessible, and affordable care [4,5,6]. Hence, nurses play a crucial role in improving the quality of patient care, which requires active participation in EBP and the application of findings in daily nurse practices [7,8].

Implementing EBP in daily nursing practices has been challenging despite its potential and reputation. Although nurses support the idea that using EBP in nursing care contributes to a better quality of care [9,10,11], many hospitals struggle with incorporating EBP into routine activities [9,10]. Previous research has indicated multifaceted barriers; for example, a lack of supporting management and willingness of nurse leaders to enhance EBP and the non-supporting of organizational and team culture [8,12,13]. In addition, nurses have mentioned a lack of time, skills, and resources, as well as difficulty reading and interpreting peer-reviewed publications, and limited research knowledge as drawbacks to implementing EBP [7].

Recently, self-efficacy in EBP (SE-EBP) has been recognized as a critical variable in predicting nurses’ preparedness for EBP [9]. Self-efficacy is defined initially as “someone’s self-perceived ability in successfully fulfilling specific tasks” [14,15]; e.g., EBP-related tasks. In this study, self-efficacy refers to nurses’ judgment of their ability to work in an evidence-based manner, and perform tasks within the EBP process. Higher self-efficacy in EBP levels correlate with a higher tendency of performing EBP [13]. In other words, nurses with higher self-efficacy levels in EBP are more likely to bring this into practice [9]. Furthermore, Chang and Crowe [16] found a strong positive correlation between self-efficacy in EBP and outcome expectancy in EBP (OE-EBP). Outcome expectancy represents the expectations of nurses that their efforts in EBP will lead to positive results [16]. Hence, focusing on increasing nurses’ self-efficacy and outcome expectancy in EBP has clear potential [17].

Hierarchical nurse leaders or managers and a work environment that facilitate EBP strengthen nurses’ beliefs in the benefits of EBP and confidence in their own EBP skills [18,19]. In such a supportive work environment, nurses can value EBP, have co-workers and physicians with an interest in EBP, have access to research databases and other resources, have time to conduct research, and receive education/skill development training and mental support for EBP [20]. In addition, positive feedback and encouragement from co-workers and nurse leaders increased nurses’ beliefs in their own capacities [10,21]. However, the association between perceived nurse leaders’ support (PLS), perceived work environment support for EBP (PWS), and nurses’ levels of self-efficacy and outcome expectancy in EBP has not yet been studied. This insight may contribute to the development of more effective implementation strategies focusing on accommodating and stimulating a culture and a productive work environment for EBP. Therefore, we performed this study answering the following research question: To what extent is perceived support for EBP from nurse leaders and nurses’ work environment associated with levels of self-efficacy and outcome expectancy in EBP among hospital nurses?

## 2. Materials and Methods

### 2.1. Study Design

A cross-sectional multicenter explorative survey study was conducted among nurses working in teaching hospital settings in the Netherlands. 

### 2.2. Setting

Hospitals were recruited via the Research and Education in Nursing (RE-Nurse) consortium. RE-Nurse includes 13 top clinical teaching hospitals in the Netherlands that pursue high-quality care through collaborations in patient care, research, education, and training. In these hospitals, EBP is an important aspect of nurses’ work. Hence, all hospitals focus on how to expand nurses’ EBP competencies and activities to improve the quality of care. Their approaches are slightly different and varied, consisting of organizing journal clubs, providing training in EBP, facilitating nurses to enroll in master studies, creating a learning network for EBP, or a combination of these activities.

### 2.3. Sample

All nurses working on clinical wards of the participating hospitals were potential participants in the survey. Nurse practitioners, outpatient clinic nurses, and research nurses were excluded as they were differently exposed to EBP than the nurses on the clinical wards. Management staff and health assistants were also excluded. The hospitals’ research departments were asked to identify wards willing to participate.

### 2.4. Data Collection

The survey consisted of two structured online self-administered questionnaires; the SE/OE in EBP questionnaire [16] and the EBP Nursing Leadership and Work Environment scales [22].

### 2.5. SE and OE-EBP Scale

We measured self-efficacy and outcome expectancy in EBP using the 37-item (subscales: self-efficacy = 29-item, outcome expectancy = 8-item) translated and adapted questionnaire of Chang and Crowe [16,23]. Self-efficacy and outcome expectancy were measured on an 11-points Likert-scale (score 0–10). Answers ranged between the extremes of “very uncertain” and “very certain”. Sum scores were separately calculated for outcome expectancy in EBP and self-efficacy in EBP and used for analysis.

### 2.6. Leadership and Work Environment Scale

Pryses’ translated 18-item questionnaire on perceived nurse leaders’ support (PLS, 10 items) and perceived work environment support for EBP (PWS, 8 items) were used [22,24]. The answers were collected on a 7-points Likert-scale (score 0–6), with answer options ranging between the extremes of “totally disagree” and “totally agree”. Sum scores were separately calculated for perceived leaders’ support and perceived work environment support and used for analysis.

Both questionnaires were translated and adapted from English to Dutch in forward and backward translation procedures, and psychometrically studied on content validity, construct validity, and reliability prior to this study (reports in preparation).

### 2.7. Covariates

Previous research has exhibited associations with the outcomes of this study. We therefore collected the following variables as covariates: working experience in years [25], educational background and student status [7], age [26], and undertaking EBP activities and collaboration in EBP [19,26]. The questionnaires also included general characteristics of the participants, e.g., ward of employment and nursing specialization, to facilitate the matching of both questionnaires.

### 2.8. Reference Items

To investigate construct validity through hypothesis testing, we added one reference item for each subscale [27]. Respondents were asked to report their overall self-efficacy in EBP, outcome expectancy in EBP, perceived support from their leaders and perceived work environment support for EBP on the same answering scale as the accompanying subscale.

### 2.9. Procedure

Eight hospitals were able to participate. The research departments of the hospitals randomly selected the wards for participation. The researchers provided the research departments with online questionnaires and invitation letters for potential participants with brief information about the study and their informed consent. Subsequently, a link to the online survey was distributed to all nurses on the selected wards via email. The research departments were unable to report the exact number of invited nurses, nor the composition of the population of nurses, which impeded determination of a response rate and investigation of selective (non-)responses. Reminder notes were sent twice. Data were anonymously collected between 1 March 2021 and 15 June 2021.

### 2.10. Ethical Considerations

The Medical Ethics Committee of the Radboud University Medical Centre confirmed that a full ethical review was not required (reference number 2020-6455). Nurses voluntarily participated and did not receive any incentives. The participants provided written informed consent in the filled-out survey. Data were separated from the general characteristics after matching and were stored and anonymously analyzed.

### 2.11. Data Analysis

Two researchers independently checked all collected data for errors (MV and PH). Complete cases were used in the analyses, i.e., when both questionnaires were matched to the same participant. Data were analyzed using Stata BE version 17.0.

We checked the internal consistency and construct validity of both questionnaires. The internal consistency was accepted with a Cronbach’s Alpha of 0.70 or above [27]. A correlation coefficient of 0.70 between the scales sum score and their reference item strongly supports construct validity [27]. Linearity of the relations between the dependent and independent variables was confirmed by plotting the data. We accepted normality of the distributions with values for Skewness and Kurtosis between −2 and 2 (Table 1).

We used mixed model analysis to explore the association between nurses’ self-efficacy in EBP with perceived leaders’ support and perceived work environment support, and the association between outcome expectancy in EBP with perceived leaders’ support and perceived work environment support. We checked for multicollinearity by identifying strong correlations (Pearson’s r > 0.70) between independent variables and the variance inflation factor (VIF). A variance inflation factor (VIF) higher than 5 indicates multicollinearity between variables within the regression models [28]. The suspected multicollinearity of age and working experience was confirmed (r = 0.95 *p* < 0.01; VIF (age) = 9.88; VIF (working experience) = 9.89). Therefore, age was omitted from the model as the authors agreed that working experience is theoretically more meaningful to one’s self-efficacy and outcome expectancy in EBP. 

Correlations between measures within hospitals were assumed, but the association models for self-efficacy and outcome expectancy with perceived support from nurses’ leaders and their work environment did not improve with a random intercept for institutions.

Backward selection of variables in the association models of self-efficacy in EBP and outcome expectancy in EBP with perceived leaders’ support, perceived work environment support, and the covariates education, student, collaborate in EBP, undertaking EBP, and working experience, resulted in two models without perceived leaders’ support. Therefore, two more models with perceived leaders’ support and without perceived work environment support were separately modeled. All four models were constructed as crude models with only the dependent variable and the independent variables. Subsequently, all covariates were added to the crude model and removed one after the other with a *p*-value higher than 0.10. The value of a random intercept for the hospitals was checked for each model.

## 3. Results

### 3.1. Sample Characteristics and Survey Findings

A total of 439 nurses from eight hospitals participated in this study, of which 306 nurses completed both questionnaires. We were unable to match 133 submitted questionnaires on self-efficacy and outcome expectancy in EBP, and 16 on perceived support for EBP scales. There were no missing items within the submitted questionnaires. The mean age of the respondents was 37.7 years (SD 12.3) and the mean working experience was 16.4 years (SD 12.4). Most respondents held a bachelor’s degree in nursing (*n* = 136) or a professional specialization (*n* = 101), e.g., in pediatrics, urology, oncology, or otherwise, and 33 respondents indicated that they were still studying. Most participants did not participate in EBP workgroups (74%) but did undertake EBP activities (68%). An overview of these characteristics is given in Table 2.

The summed scores of the subscales self-efficacy in EBP and outcome expectancy in EBP were 176.3 (SD 47.01, Max. = 290) and 56.8 (SD 12.4, Max. = 100), respectively. For perceived leadership support, the summed score was 32.2 (SD 13.5, Max. = 70) and for perceived work environment support it was 29.9 (SD 7.49, Max. = 80). The summed scores for all participants and per hospital are shown in Table 2.

### 3.2. Perceived Support from Work Environment

Perceived support from nurses’ work environment showed stronger associations with self-efficacy and outcome expectancy in EBP than perceived support from nurses’ leaders (Table 3 and Table 4). The individual regression models showed that nurses with higher self-efficacy in EBP scores experienced more support for EBP from their work environment and leaders.

Self-efficacy in EBP is associated with the perceived support from nurses’ work environment (B = 1.25; SE = 0.31; 95%CI [0.65–1.85]). A nurse whose perceived work environment support increases by one unit on the measurement scale is likely to have an increased self-efficacy in EBP between the 0.65 and 1.85 units. Outcome expectancy in EBP is associated with perceived support from nurses’ Work Environment (B = 0.35; SE = 0.08; 95%CI [0.19–0.51]). Again, a nurse who perceives work environment support for EBP, one unit higher than before, probably has a higher outcome expectancy in EBP between 0.19 and 0.51 units on that measurement scale.

### 3.3. Perceived Support from Nurses’ Leaders

Perceived support from nurses’ leaders was not significant within models with perceived support from nurses’ work environments, and therefore, this was modeled separately (Table 3 and Table 4). Perceived support from nurses’ leaders was then associated with self-efficacy (B = 0.39; SE = 0.17; 95%CI [0.05–0.74]) and outcome expectancy in EBP (B = 0.13; SE = 0.05; 95%CI [0.03–0.22]). Nurses who perceive one unit more of support in EBP from their managers are likely to have a 0.39 higher self-efficacy in EBP and a 0.13 higher outcome expectancy.

### 3.4. Covariates in the Models

All four models contain the covariates educational level, undertaking EBP activities, and work experience. Being a student is only present in the models for perceived leaders’ support for EBP. Undertaking EBP activities has clearly the strongest and most stimulating influence on nurses’ self-efficacy in EBP and their outcome expectancy. Additionally, educational level positively influences nurses’ self-efficacy and outcome expectancy in EBP.

At the time of study, nurses with a longer work experience had lower self-efficacy and outcome expectancy in EBP. Additionally, and only for perceived leaders’ support for EBP, being a student negatively affected self-efficacy and outcome expectancy in EBP. The corresponding regression coefficients are shown in Table 3 and Table 4.

Most respondents did not actively collaborate in EBP working groups, and some of them noted that their EBP working group was suspended due to COVID-19 priorities. Within our models, actively participating in a EBP working group did not have a significant influence.

## 4. Discussion

This study showed that EBP-supportive leaders and work environments positively contribute to nurses’ self-efficacy and outcome expectancy in EBP, along with the covariates of undertaking EBP activities and educational level. This is consistent with Bandura’s Social Cognitive Theory [15]. When applied to our study, self-efficacy developed through positive experiences when undertaking EBP (mastery experience) and receiving positive feedback from fellow nurses and physicians (verbal persuasion) [15]. The level of education and undertaking EBP activities were also strongly present within our models of self-efficacy and outcome expectancy in EBP. Yoo et al. [19] reported that participating in EBP activities contributed to increased self-efficacy and outcome expectancy; exposure to EBP gives confidence and underlines its importance. Nurses experience their own ability to fulfill EBP activities, and that increases their EBP-skills [19,26]. Further training is indispensable, which illustrates the importance of active involvement in the EBP process as a part of nursing practice. Moreover, support for EBP from colleagues, physicians, and the organization appears of greater importance than that of nurse leaders. It may indicate that facilitating EBP is required at the level of an organization, and that encouraging and awarding EBP behavior is of greater value when it comes from fellow nurses and physicians instead of nurse managers. Additionally, in practice, a supportive work environment and supportive leaders may be inextricably linked; for example, in self-directed teams.

Our questionnaires also provided insight on the levels of self-efficacy and outcome expectancy in EBP, and the perceived support from nurses’ leaders and work environment. Self-efficacy in EBP and outcome expectancy in EBP were rated at 60% and 57% of the maximum scores, respectively. The perceived support for EBP from nurses’ leaders and their work environments reached 46% and 37% of the maximum scores, respectively. This suggests that nurses lacked both a supportive work environment and support from their leaders, despite being part of an EBP-orientated organization.

We found negative associations of work experience with self-efficacy and outcome expectancy in EBP. Nurses with less work experience have probably graduated more recently in a curriculum that has developed alongside the growing need for EBP. Due to changes to the school curriculum, recently graduated nurses have less work experience, but receive more education in EBP [26].

We included 306 nurses from randomly chosen wards, consisting of EBP working groups and EBP coaching programs, from eight top clinical teaching hospitals in the Netherlands. The other five hospitals from the RE-Nurse consortium found less participation opportunities due to the COVID-related burden on their nurses. We also were unable to determine the exact number of invited nurses or the composition of the population of nurses. Therefore, it was not possible to determine a response rate or investigate selective (non-)responses. Additionally, it must be considered that data collection took place between two contamination peaks of COVID-19, which was likely to have suppressed the response and ability of nurses to work together in EBP workgroups or undertake EBP activities.

Our translated questionnaires showed adequate validity and reliability. Cronbach’s Alpha for internal consistency met the cut-off points for all subscales (Table 1). Pearson’s correlation coefficients supported the construct validity of the three questionnaires; the perceived work environment support scale did not reach the 0.70 cut-off (Table 1).

A random intercept for the various hospitals did not improve the regression models, meaning that presupposed differences between hospitals were not present in our data. This may have occurred because of the similarity between the goals of these hospitals and communalities in the support of EBP and research in nursing. The assumed conformity among the hospitals in this study may be a limitation in generalizability. The relationships between nurses’ self-efficacy and outcome expectancy with their perceived support from nurses’ leaders and their work environment may not differ. However, the levels of each of these variables are likely to vary between the various healthcare institutions.

The crude models containing both perceived leaders’ support and perceived work environment support show small, negative, and not significant coefficients for perceived leaders’ support for EBP. Similar values appeared before the backward selection procedure, which resulted in the removal of perceived leaders’ support for EBP from the model. One item on Pryse’s Work Environment scale concerns the manager’s role in providing access to resources. Additionally, as nurses’ leaders represent their organizations, the influence of nurses’ leaders seems overruled by the perceived support of the work environment in the models for self-efficacy and outcome expectancy in EBP.

Some limitations apply when interpreting the results of this study. Since we were unable to investigate selective non-response, our respondents may represent nurses with more positive attitudes to, or interests in, EBP. Additionally, our data collection software did not capture incomplete questionnaires because informed consent was only given when submitting the questionnaire. Furthermore, all our respondents work at top clinical teaching hospitals in the Netherlands that share a special interest in nursing research. This may positively affect nurses’ perceived support for EBP, and their own self-efficacy and outcome expectancy scores compared to nurses in less distinctive hospitals or healthcare organizations.

## 5. Conclusions

The findings of this study put the importance of support for EBP from nurse leaders and the nurses’ work environments into perspective. The findings of this study signified the importance of support from nurses’ leadership, and were relative to the support for EBP that comes from the nurses’ work environment. The support of a manager was important if the work environment was not considered. Support from the work environment, or the lack thereof, blurs the effects of a supportive manager. Theoretically, self-efficacy and outcome expectancy precede behavior [15]. Thus, as an answer to the call for appropriate, accessible, and affordable care [4,5,6], nurses need to develop their self-efficacy and outcome expectancy in EBP. Supportive leaders need to pay specific attention to building a supportive work environment. Additionally, supporting and encouraging peers in EBP should be part of the curricula of nursing studies and professional training, so that nurses develop fundamental self-efficacy and outcome expectancy in EBP during their studies.

## 6. Unanswered Questions and Further Research

It is necessary to investigate whether participation in EBP workgroups with colleagues really has little influence on nurses’ self-efficacy and outcome expectations in EBP, as collaboration is ascribed as a source of self-efficacy. To measure the influence of collaboration with peers in more detail, it may be necessary to extend the questionnaire for perceived support from nurses’ work environment or develop a specific subscale.

Furthermore, we are curious about the development of self-efficacy and outcome expectancy over time in relation to undertaking EBP activities and interventions that cause this to happen.

## Figures and Tables

**Table 1 ijerph-19-14422-t001:** Distribution characteristics and internal consistency.

Scale (N Items)	Mean (SD)	Skewness (SE)	Kurtosis (SE)	Cronbach’s Alpha	Pearson Correlation ^5^ r (*p*-Value)
SE-EBP ^1^ (29)	176.27 (47.01)	−0.71 (0.14)	0.92 (0.28)	0.98	0.83 (*p* < 0.001)
OE-EBP ^2^ (10)	56.75 (12.40)	−0.93 (0.14)	1.6 (0.28)	0.95	0.76 (*p* < 0.001)
PLS ^3^ (10)	32.20 (13.49)	−0.38 (0.14)	−0.19 (0.28)	0.96	0.83 (*p* < 0.001)
PWS ^4^ (8)	29.92 (7.49)	−0.36 (0.14)	0.29 (0.28)	0.80	0.67 (*p* < 0.001)

^1^: Self-efficacy in EBP. ^2^: Outcome-expectancy in EBP. ^3^: Perceived support from nurses’ leaders. ^4^: Perceived work environment support. ^5^: Correlation between the scales sum score and the accompanying reference item.

**Table 2 ijerph-19-14422-t002:** Sample characteristics.

Hospitals 1–8	Totals	H1	H2	H3	H4	H5	H6	H7	H8
**Participants**									
SE and OE Scale	439	67	27	12	134	16	82	95	6
LS and WS Scale	322	47	21	11	93	12	64	68	6
Matched scales	306	41	19	11	92	11	61	65	6
**Educational level**									
Vocational Nurse (NLQF4)	49	4	1	-	19	3	9	13	-
Professional Nurse	1	-	-	-	1	-	-	-	-
Bachelor Nurse (NLQF6)	136	17	18	9	36	3	24	23	6
Specialized Nurse (NLQF6)	101	15	-	1	30	5	24	26	-
Nurse Specialist (NLQF7)	7	1	-	-	4	-	1	1	-
Nurse Academic (NLQF8)	12	4	-	1	2	-	3	2	-
**Student**									
No	273	33	15	10	85	10	55	59	6
Yes	33	8	4	1	7	1	6	6	-
**Collaboration in EBP**									
None	227	34	11	2	74	8	46	47	5
Care Discussion	5	2	-	-	1	-	1	1	-
EBP Workgroup	69	5	8	7	16	3	12	17	1
Journal Club	5	-	-	2	1	-	2	-	-
**Undertaking EBP activities**									
None	97	12	2	3	34	2	20	22	2
Yes	209	29	17	8	58	9	41	43	4
**Variable**	**Mean (SD)**								
Age	37.67 (12.34)	38.17 (13.74)	30.37 (7.40)	34.55 (8.35)	38.92 (12.09)	37.18 (13.57)	35.82 (10.65)	40.69 (14.00)	30.83 (8.26)
Working Experience	16.41 (12.43)	16.46 (13.28)	9.74 (8.25)	12.36 (8.30)	17.54 (12.15)	18.27 (13.90)	14.35 (11.08)	19.68 (14.07)	9.25 (7.87)
SE-EBP Max. = 290	176.27 (47.01)	182.56 (42.69)	177.68 (23.89)	184.82 (32.11)	173.26 (45.60)	167.82 (26.14)	178.54 (52.39)	172.29 (57.15)	195.00 (25.15)
OE-EBP Max. = 100	56.75 (12.40)	57.39 (13.17)	57.37 (7.59)	60.55 (7.88)	54.66 (12.69)	55.18 (11.43)	58.59 (11.20)	56.78 (14.72)	59.17 (3.97)
PLS Max. = 70	32.20 (13.49)	29.76 (15.78)	37.21 (12.03)	36.27 (12.59)	32.57 (12.23)	33.73 (12.81)	26.43 (14.13)	36.15 (11.17)	33.00 (20.74)
PWS Max. = 80	29.92 (7.49)	26.56 (9.34)	32.05 (5.66)	34.18 (5.55)	30.83 (7.51)	32.73 (5.83)	28.28 (5.72)	30.75 (7.57)	27.00 (9.38)

**Table 3 ijerph-19-14422-t003:** Linear mixed model analysis of self-efficacy in EBP with nurses’ perceived work environment support and perceived nurse leaders’ support.

	Crude Model					Model with Nurses’ Perceived Work Environment Support 1					Model with Perceived Support from Nurses’ Leaders 2				
	Beta	B (SE)	95%-CI	z	*p*-Value	Beta	B (SE)	95%-CI	z	*p*-Value	Beta	B (SE)	95%-CI	z	*p*-Value
Cons.	.	128.83 (10.95)	107.38–150.29	11.77	<0.001		123.08 (10.91)	101.61–144.55	11.28	<0.001	.	146.13 (8.73)	128.95–163.30	16.74	<0.001
PLS ^3^	−0.08	−0.29 (0.26)	−0.78	−1.08	0.28						0.11	0.39 (0.17)	0.05–0.74	2.27	0.024
PWS ^4^	0.29	1.88 (0.47)	0.95–2.80	3.99	<0.001	0.20	1.25 (0.31)	0.65–1.85	4.10	<0.001					
Educational level						0.25	13.09 (2.56)	8.04–18.13	5.11	<0.001	0.29	13.39 (2.63)	8.21–18.57	5.08	<0.001
Undertaking EBP activities						0.22	22.39 (5.31)	11.95–32.83	4.22	<0.001	0.51	23.88 (5.40)	8.21–18.57	5.08	<0.001
Work experience						−0.27	−1.03 (0.19)	−1.41–−0.65	−5.34	<0.001	−0.02	−1.02 (0.20)	−1.41–−0.63	−5.15	<0.001

^1^: Self-efficacy in EBP and nurses’ perceived work environment support (PWS), corrected for educational level, undertaking EBP activities, and work experience. Perceived support from nurses’ leaders (PLS) and the covariates student and collaborating in EBP were excluded due to the modeling procedure. ^2^: Self-efficacy in EBP and perceived support from nurses’ leaders (PLS), corrected for educational level, undertaking EBP activities, and work experience. The covariates ‘student’ and ‘collaborating in EBP’ were excluded due to the modeling procedure. ^3^: Perceived support from nurses’ leaders (PLS). ^4^: Nurses’ perceived work environment support (PWS).

**Table 4 ijerph-19-14422-t004:** Linear mixed model analysis of outcome expectancy in EBP with nurses’ perceived work environment support and perceived nurse leaders’ support.

	Crude Model					Model with Nurses’ Perceived Work Environment Support ^1^					Model with Perceived Support from Nurses’ Leaders ^2^				
	Beta	B (SE)	95%-CI	z	*p*-Value	Beta	B (SE)	95%-CI	z	*p*-Value	Beta	B (SE)	95%-CI	z	*p*-Value
Constante	.	44.42 (2.91)	38.70–50.14	15.28	<0.001	.	42.62 (2.91)	36.91–48.32	14.64	<0.001	.	48.69 (2.35)	44.06–53.32	20.69	<0.001
PLS ^3^	−0.06	−0.05 (0.07)	−0.19–0.08	−0.78	0.439						0.14	0.13 (0.05)	0.03–0.22	2.69	0.008
PWS ^4^	0.28	0.47 (0.12)	0.22–0.71	3.75	<0.001	0.21	0.35 (0.08)	0.19–0.51	4.35	<0.001					
Educational level						0.27	3.67 (0.72)	2.26–5.07	5.12	<0.001	0.28	3.79 (0.75)	2.32–5.26	5.08	<0.001
Student						−0.12	−4.78 (2.10)	−8.90–−0.67	−2.28	.023	−0.13	−5.04 (2.17)	−9.31–−0.76	−2.32	0.021
Undertaking EBP activities						0.20	5.37 (1.42)	2.59–8.15	3.79	<0.001	0.22	5.81 (1.45)	2.95–8.68	4.00	<0.001
Work experience						−0.27	−0.27 (0.05)	−0.38–−0.17	−5.24	<0.001	−0.27	−0.27 (05)	−0.38–−0.17	−5.05	<0.001

^1^: Outcome expectancy in EBP and nurses’ perceived work environment support (PWS), corrected for educational level, student, undertaking EBP activities, and work experience. Perceived support from nurses’ leaders (PLS) and the covariate collaborating in EBP were excluded due to the modeling procedure. ^2^: Outcome expectancy in EBP and perceived support from nurses’ leaders (PLS), corrected for educational level, student, undertaking EBP activities, and work experience. The covariate ‘collaborating in EBP’ was excluded due to the modeling procedure. ^3^: Perceived support from nurses’ leaders (PLS) ^4^: Nurses’ perceived work environment support (PWS).

## Data Availability

The questionnaires and datasets generated and/or analyzed during the current study are not publicly available due to the nature of the data but are available from the corresponding author on reasonable request.

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
