# Peer review of "Nurse Leadership and Work Environment Association with Outcome Expectancy and Self-Efficacy in Evidence-Based Practice among Hospital Nurses in The Netherlands: A Cross-Sectional Study"

_ijerph, 2022, doi:10.3390/ijerph192114422_

Round 1
Reviewer 1 Report
This is a study dealing with a very intersting and innovative issue of nursing research.
I have some minor suggestions for revision before a publication is warranted.
1. In the manuscript you write scale names (self-efficacy etc) and covariates (Age, Working Experience etc.) in capital letters (but I think not fully consistent). This is rather unsusal to my experience. Please consider to write them in lower case letters.
2. p.4. You formulate a research question but you refer so much to theory and other studies that it shoul be posssible to deduce hypotheses. Please consider to formulate hypotheses, test them, give answers if they are supported or not in the results section.
3. Methods
3.1 Missing: Are there missings in your data? How you dealth with that issue, also because you used sum scores
3.2 You state that a method paper on this study is in preparation. Internal consistency estiamtes are reported. However, I would expect that give some information on construct validity. Is it possible to conducte a (multilevel adjusted) confirmatory factor analysis revealing that you have distinct constructs?
3.3 What is the participation rate in your sample? (439/Total number of nurses)
4. Results
4.1 (a) Table 3-6 header should be "Results of Lineare Mixed Modelling on Relationships between ... and ..." and (b) in this header present [I] the independent variable(s) and [II] the dependent variable, and (c) p cannot be "0", therefore change p = .000 into p < .001.
4.2 general comment: please add notes below all the tables explaining the abbrevations of variables
4.3 For me it reamins unclear why there are 4 tables? As I understand you have to ouctomes (self-efficacy and outcome expectancy) => in this sense, there should be 2 two tables (one for each outcome), of course you could include different models here
5. Could you please add some information on the limitations of your study or introduce the respective paragraph with such a statement "When interpreting the result of ours study the following limitations have to be considered."
Author Response
Dear editors and reviewers,
On behalf of my coauthors, I thank you for your remarks and feedback on our manuscript for consideration for the International Journal of Environmental Research and Public Health. We have revised our manuscript entitled “Nurse leadership and work environment association with outcome expectancy and self-efficacy in Evidence-Based Practice among hospital nurses in The Netherlands: a cross-sectional study” according to your feedback.
Please find our detailed manuscript with tracked changes and our responses to the remarks and feedback attached. The most important changes and additions are:
- Consistently used lower case letters for variables and covariates.
- Added a column on Pearson’s correlations to report about construct validity.
- Added information about our inability to match 133 SE/OE in EBP questionnaires and 16 PWS/PLS questionnaires and the absence of missing items in submitted questionnaires.
- Merged tables 3-6 into tables 3 and 4 with accompanying titles and legends.
- Removed misplaced and added missing commas.
- Changed () into [] when reporting confidence intervals in text.
- Added an alinea on reference items in the questionnaires in the methods section. Reference items are used to investigate the support for construct validity.
- Added an alinea on limitations when interpreting the results of this study.
We have chosen not to formulate distinct hypothesis because information about the magnitude of the relations between the variables and covariates was not available. To formulate these reflective, knowing the results of our study seems not appropriate at this point. Also, reporting the confirmative factor analysis to support construct validity for four subscales within this manuscript is too complex. We are preparing two manuscripts on psychometric properties of each of the questionnaires in this study. However, we have added information about the hypothesis testing we also performed within this study, but which was not part of the original manuscript.
This study is not subjected to ethical review according to Dutch law (reference number 2020-6455, Medical Ethics Committee of the Radboud Medical Centre in The Netherlands).
Our revised manuscript contains 3345 words, excluding tables and the reference list. The summary consists of 198 words.
The content of this revised manuscript has not been previously published and is not under consideration elsewhere. There is no conflict of interest for the individual authors.
All authors have approved and confirmed with submission of the manuscript to the special issue “Culture of Evidence-Based Practice and Quality Improvement in Nursing” of the “International Journal of Environmental Research and Public Health”.
Thank you for your consideration. We look forward to your response.
Kind regards,
Peter Hoegen

Reviewer 2 Report
Very interesting and worthy of further discussion and research as well. There were minor language usage in abstract( af-fordable, show). Leadership and nursing is paramount especially during global shortage of medical and support staff. Thank you for sharing.
Line 34-Grow ( language/vacab edit)
Line 43 Definition
Line 88 Perhaps elaborate on definition of Cross section . In sample are there other criteria, if so may list or add to description. Such as years of service or type of nursing degrees ...
Excellent discussion points and connections with graphics, visuals
Line 326 Edit language
Overall, very comprehensive and explained/ researched.
Author Response

(The authors gave the same response as above.)
